# Lung Cancer Organoids: The Rough Path to Personalized Medicine

**DOI:** 10.3390/cancers14153703

**Published:** 2022-07-29

**Authors:** Rachele Rossi, Maria Laura De Angelis, Eljona Xhelili, Giovanni Sette, Adriana Eramo, Ruggero De Maria, Ursula Cesta Incani, Federica Francescangeli, Ann Zeuner

**Affiliations:** 1Department of Oncology and Molecular Medicine, Istituto Superiore di Sanità, Viale Regina Elena 299, 00161 Rome, Italy; rachele.rossi@iss.it (R.R.); marialaura.deangelis@iss.it (M.L.D.A.); giovanni.sette@iss.it (G.S.); adriana.eramo@iss.it (A.E.); federica.francescangeli@iss.it (F.F.); 2Department of Surgical Sciences, Policlinico Umberto I, Sapienza University of Rome, Viale del Policlinico 155, 00161 Rome, Italy; xhelili.1890473@studenti.uniroma1.it; 3Institute of General Pathology, Università Cattolica del Sacro Cuore, Largo Francesco Vito 1, 00168 Rome, Italy; ruggero.demaria@unicatt.it; 4Fondazione Policlinico Universitario A. Gemelli IRCCS, 00168 Rome, Italy; 5Division of Oncology, University and Hospital Trust of Verona (AOUI), Piazzale Ludovico Antonio Scuro 10, 37134 Verona, Italy; ursula.cestaincani@aovr.veneto.it

**Keywords:** lung cancer, organoids, preclinical models, drug testing, targeted therapy, personalized medicine

## Abstract

**Simple Summary:**

Lung cancer is very difficult to cure, especially when it has spread to other parts of the body. One of the main reasons that delay the identification of effective therapies is the complexity of lung cancer cells, which can be very different among individual patients. Organoids are small aggregates of tumor cells that are generated from lung tumors and used in cancer research laboratories to study the features of tumor cells. Organoids have unique properties, as they reproduce many characteristics of the tumor derived from each specific patient. Due to their capacity to reproduce individual tumor features in the laboratory setting, organoids are an excellent system to study lung cancer and to identify functional therapies. This review summarizes the challenges encountered by researchers in the field of lung cancer organoids and describes how the advances in organoid technology may allow the future development of individualized therapies for lung cancer patients.

**Abstract:**

Lung cancer is the leading cause of cancer death worldwide. Despite significant advances in research and therapy, a dismal 5-year survival rate of only 10–20% urges the development of reliable preclinical models and effective therapeutic tools. Lung cancer is characterized by a high degree of heterogeneity in its histology, a genomic landscape, and response to therapies that has been traditionally difficult to reproduce in preclinical models. However, the advent of three-dimensional culture technologies has opened new perspectives to recapitulate in vitro individualized tumor features and to anticipate treatment efficacy. The generation of lung cancer organoids (LCOs) has encountered greater challenges as compared to organoids derived from other tumors. In the last two years, many efforts have been dedicated to optimizing LCO-based platforms, resulting in improved rates of LCO production, purity, culture timing, and long-term expansion. However, due to the complexity of lung cancer, further advances are required in order to meet clinical needs. Here, we discuss the evolution of LCO technology and the use of LCOs in basic and translational lung cancer research. Although the field of LCOs is still in its infancy, its prospective development will likely lead to new strategies for drug testing and biomarker identification, thus allowing a more personalized therapeutic approach for lung cancer patients.

## 1. Introduction

According to the GLOBOCAN database, lung cancer has been the second most commonly diagnosed cancer and the leading cause of cancer death globally in 2020, with an estimated 1.8 million deaths [1]. The 5-year survival of lung cancer patients after diagnosis is 10–20% in most countries. Early detection allows for the adoption of potentially curative treatment approaches, thus providing higher survival rates. However, three quarters of patients are diagnosed when the disease has spread to lymph nodes and/or other organs and is virtually incurable [2]. Lung cancer is broadly classified into two major categories: small-cell lung cancer (SCLC) and non-small-cell lung cancer (NSCLC). NSCLC is further divided into lung adenocarcinoma (LUAD), squamous cell carcinoma (LUSC) and large-cell carcinoma (LCC). Ever-improving molecular subtyping techniques are used to guide treatment strategies and to allow a better prediction of clinical trajectories. Therapeutic regimens for lung cancer are established according to histologic subtypes and genetic profiling. First-line targeted therapies are the current standard of care treatment for NSCLC patients with actionable mutations, such as those in EGFR, ROS1, ALK, NTRK1, BRAFV600E, and RET. KRAS inhibitors are the preferred choice for patients with the KRASG12C mutation, while patients with wild-type driver genes typically receive immune checkpoint inhibitors with or without platinum-containing chemotherapy [3]. Treatment options for lung cancer are continuously increasing, but they usually provide modest benefits in terms of survival. The absence of reliable biomarkers of drug efficacy together with the development of treatment resistance by cancer cells contribute to limit the effectiveness of current therapies. The identification of individual biomarkers will allow more personalized therapeutic approaches for lung cancer patients, also avoiding the unnecessary treatment of non-responders.

Lung cancer research during the past few decades has relied mainly on established cell lines. These have many limitations as they are relatively homogeneous, they lack interactions with the tumor microenvironment, and they may accumulate genetic and epigenetic aberrations upon in vitro culture. Although in some cases cell lines may mimic clinical drug responses (i.e., cell lines with distinctive EGFR mutations show a different sensitivity to tyrosine kinase inhibitors) [4], most experimental therapies tested on established cell lines failed phase III clinical trials [5]. More recently, preclinical lung cancer models have progressed to include patient-derived xenografts (PDXs) and lung cancer organoids (LCOs). Both PDXs and LCOs reproduce the genetic and histological heterogeneity of parental tumors, recapitulating lung cancer pathogenesis more faithfully as compared to cell lines. While both models provide unprecedented opportunities to lung cancer research, they are associated with important limitations that delay their use for drug discovery and clinical treatment decisions. PDXs are expensive, time-consuming, and unsuitable for large-scale drug screening experiments. Organoids are more versatile, as they can be adapted to high-throughput drug screening platforms—to study cancer-microenvironment interactions—and to gene-editing techniques. However, the complexity and heterogeneity of lung tumors is holding back the use of LCOs in preclinical and clinical lung cancer research. Substantial efforts are needed to understand LCO culture requirements and to develop improved techniques for LCO-based drug testing. Nevertheless, the implications of organoid development for lung cancer treatment are huge and the field is gaining momentum. In this review, we discuss the technical evolution, development, and applications of LCOs in preclinical and clinical research. Starting with a brief history of normal and malignant lung organoids, we will focus on the challenges encountered in LCO development, on the possible solutions, and on the use of LCOs in basic and translational lung cancer research. Future developments of LCOs for drug testing and personalized therapies are also discussed.

## 2. A Brief History of Normal and Neoplastic Lung Organoids

For decades, preclinical lung cancer research has relied on cell lines grown in two-dimensional (2D) cultures. These, however, have a limited ability to reproduce tumor complexity and drug responses due to the inability to reproduce tumor heterogeneity, the absence of a three-dimensional (3D) architecture, and the progressive accumulation of genetic mutations (the so-called “genetic drift”) [6]. Lung cancer cell lines have been used to produce tumor spheroids, which have been used for multiple studies including analyses of molecular crosstalk in a tumor microenvironment (TME), identification of cell–to-cell interactions, and drug screening [7]. Cell-line-based spheroids provide drug responses more similar to those of tumors in vivo as compared to 2D cultures, as their multi-layer structure with hypoxic, senescent, and necrotic areas can, to some degree, mimic the solid tumor architecture [8]. Moreover, the cheapness and reproducibility of cell-line-derived spheroids are advantageous for testing the physicochemical properties of drug candidates, such as drug penetration [7]. Cell lines, including lung adenocarcinoma Calu-3 cells, can also be cultured in Matrigel to stimulate cell polarization and differentiation [6]. However, cell-line-based models are, overall, of limited utility for preclinical drug testing and unsuitable for personalized therapeutic approaches. Patient-derived lung cancer spheroids were introduced in 2008 by Eramo and coworkers, providing a personalized 3D model able to generate xenografts that recapitulated the histology of parental tumors [9]. Patient-derived lung cancer spheroids were used for in vitro and in vivo studies by our group and others [10,11,12,13,14], providing a reliable preclinical model for drug testing and molecular analyses. The advent of patient-derived organoids (PDOs) provided a further improvement to tumor modeling in terms of structural complexity, cell differentiation, heterogeneity, and physiological function [15]. The generation of organoids from normal epithelial cells and from lung cancer cells proceeded in parallel for some time, as illustrated in Figure 1. However, as discussed in detail in the following section, the field of LCOs was shaken by the discovery that normal lung organoids tend to prevail over LCOs during the establishment of organoid cultures from primary lung tumors [16,17]. Since then, it was recognized that LCO generation required more sophisticated protocols and more careful validations than previously thought. Table 1 reports the studies that contributed to the main advances in LCO development, starting with a “pre-organoid era” that includes the generation of spheroid cultures from lung tumors. The first self-renewing 3D cultures of lung cancer cells consisted of tumor spheroids enriched in tumor-initiating cells derived from surgical specimens of both SCLC and NSCLC [9]. Patient-derived spheroids were used in several attempts to generate personalized models for lung cancer [14,18,19,20,21]. However, the application of lung cancer spheroids for patient prognosis and treatment choice remains limited, while they remain useful for the discovery of molecular targets and for the evaluation of drug candidates [7]. Subsequently, NSCLC spheroid generation from tumor tissues or pleural effusion was performed by Endo et al. using Matrigel [22], moving close to organoid culture conditions. The first attempt to generate lung 3D cultures in a semi-structured environment employed human respiratory epithelial cells isolated from nasal polyps and maintained in a serum-free medium within collagen lattices. This method produced tubular structures containing cuboidal-shaped polarized cells, ciliated cells, secretory cells, and undifferentiated cells, with epithelial cells capable of contraction [23]. 3D culture systems originated from normal lung stem cells (tracheospheres and alveolospheres) were developed years later [24,25], preparing the terrain for lung organoid development. The first generation of normal lung organoids took advantage of human induced pluripotent stem cells (iPSC) [26,27,28] and was shortly followed by organoids derived from adult lung primary cells [29] and from embryonic lung epithelial cells [30]. Normal airway organoids were used to recapitulate lung development and to model lung disease, such as viral infection [26] or cystic fibrosis [28,29,30,31].

The first appearance of LCOs in the scientific literature dates back to 2017, once among a broad multi-tumor approach to individualized cancer models [32] and within a small-scale generation of tumoroids from colorectal and lung cancers [33]. In both cases, the number of processed tumors and LCO generation was small (*n* = 1/2 and *n* = 3/3 respectively) and the second study reported problems in LCO propagation [33]. The systematic establishment of LCOs from lung cancer had to wait until 2019 (with preprint publication in December 2018), when the Clevers group described the generation of 19 LCO lines, as well as airway organoids derived from peritumoral tissue and from cystic fibrosis patients [34]. LCOs generated by Sachs and coworkers retained the histopathology and mutation profile of original tumors and were amenable both to small-scale drug-screening experiments and to orthotopic transplantation [34]. Sachs and coworkers also proposed a solution to the main challenge in LCO generation, i.e., the overgrowth of normal airway organoids. To overcome this obstacle, the authors performed LCO selection with the MDM2 antagonist Nutlin-3a, as this would kill TP53 wild-type cells while sparing cells with mutant TP53 [34]. Nutlin-3a selection allowed the generation of virtually pure LCO lines from primary lung tumors and was subsequently adopted by other investigators (Table 1). However, as discussed in the following paragraph, Nutlin-3a selection presents some drawbacks and other methods used to avoid LCO contamination by normal airway organoids that are under investigation (Table 2). LCOs were also developed from extrapulmonary sites, such as metastases (lymph nodes, brain, and bone marrow) and malignant effusions [35,36], recapitulating tumor-like histology and drug responses. Shortly after those pioneer studies, an increasing number of publications described the use of LCOs to investigate drug responses [35,37,38,39,40,41,42] and to test immunotherapeutic strategies in co-cultures with autologous T cells [43,44,45].

## 3. Problems and Solutions in the Generation of Lung Cancer Organoids

The generation of LCOs has been more difficult as compared to organoids from other tumors and still represents a challenge. The main problems encountered during LCO generation consist of contamination by normal airway organoids, low success rate of culture establishment, low culture yields, inadequacy of media formulations, and lengthy times incompatible with clinical needs (Table 2). As long as the variables influencing LCO production started to be elucidated, possible solutions have been proposed for each problem (Table 2). Examples of variables influencing the success of LCO culture establishment include the size of tumor specimens, the amount of vital cancer cells in surgical resections, and multiple parameters associated with tissue processing (i.e., time, temperature, and digestion protocol). Furthermore, the optimal culture conditions for LCOs are yet to be defined, taking into account that different genetic backgrounds are probably related to different growth factor requirements, as observed for colorectal cancer organoids [58]. A detailed protocol for LCO production from LUAD samples has been published in 2021, reporting a success rate of 80% [53], while LCO generation from LUSC is reportedly more difficult and awaits optimization.

The first studies that systematically addressed the challenges encountered in LCO generation recognized that the majority of LCOs derived from primary tumors were overgrown by normal airway organoids at an estimated rate around 65% [16,17]. This finding is in line with our previous studies showing overgrowth of non-tumor lung cells also in patient’s tumor-derived primary cultures established with a totally different approach [59]. Altogether, these observations suggest that overgrowth of normal cells is a general behavior of lung cells not restricted to 3D organoid cultures, further highlighting the need of both a selection method for preventing normal organoid growth and a multi-step validation of LCOs in order to confirm their neoplastic origin [59]. In order to inhibit the overgrowth of normal lung organoids, Dijkstra et al. and Sachs et al. introduced the use of the MDM2 inhibitor Nutlin-3a. Nutlin-3a kills TP53 wild-type cells while sparing TP53 mutated cancer cells [34,44]. However, this method counter-selects a consistent portion of LCOs derived from TP53 wild-type patients, who represent around 50% of cases [60]. Moreover, Nutlin-3a selection may promote the growth of peritumoral TP53-mutated airway epithelial cells and may cause toxicity problems in long-term cultures. Multi-step LCO validations have also been proposed in order to verify culture purity. These include histomorphology, genetic analyses, immunohistochemistry, and xenograft formation in immunocompromised mice [16,17,40] (Table 3). Other methods aimed at avoiding the competitive growth of normal lung epithelial cells include the use of a medium without the factors required for the growth of normal lung organoids [39], hand-picking of tumor organoids [16,53], and derivation of cancer cells from extrapulmonary sources, such as metastases, pleural effusions, or PDXs [35,36,38]. LCO derivation from extrapulmonary metastatic sites or pleural effusion should avoid the competitive growth of normal lung epithelial cells. Indeed, some authors have successfully reported LCO generation from extrapulmonary metastases [33,34,35,42] or from malignant effusion samples [35,36,42,57]. LCO generation from lung cancer metastases has been reported to have a higher rate of success as compared to intrapulmonary samples [16] but is generally hindered by the small size of biopsies. Malignant effusion samples have been reported to give rise to LCOs, although the number of studies is still low [35,36,42]. Kim et al. produced 77 LCO lines starting from effusion samples (pleural, pericardial, or ascites), showing that they reproduced the genetic features of advanced LUAD and they were amenable for predictive studies [35]. However, malignant effusion samples are available only from advanced-stage patients. Thus, an optimization of protocols for LCO derivation from primary tumors remains essential for effective clinical application to both early-stage and late-stage patients. LCOs have also been generated from PDXs established from NSCLC [17,39] or SCLC [38]. PDX-derived LCOs showed drug responses comparable to those of PDXs [17,39], supporting the hypothesis that organoids are a good substitute for PDXs in preclinical drug screening and in personalized treatment strategies. LCO generation from PDXs has the advantage of avoiding the overgrowth of normal lung organoids, as normal lung epithelial cells should not grow in subcutaneous xenografts. Moreover, PDXs were reported to generate organoids more easily as compared to primary tumors [17]. However, an important outgrowth of normal epithelial cells or of murine cells was reported [17], indicating that further adjustments to this method are needed.

## 4. Preclinical Applications of Lung Cancer Organoids

### 4.1. LCOs in Basic Research: A Useful Tool to Understand Lung Cancer Biology

The unique properties of tumor organoids make them an excellent model for tumor biology studies. Organoids combine the experimental advantages of in vitro systems (including long term expansion, cryopreservation, and genetic manipulation) with the 3D architecture and differentiation of in vivo models. Organoids from several tumor types have been successfully used to study key aspects of cancer biology, such as self-renewal, drug resistance, heterogeneity, and oncogenic transformation [61]. Moreover, organoids allow the investigation of tumor compartments traditionally difficult to reproduce in vitro, such as circulating tumor cells [62]. Organoids from gastrointestinal tissues have been used to model tumor initiation and evolution, providing a proof of principle that the oncogenic process can be recapitulated in vitro in a tissue-specific fashion [63,64,65,66]. Neoplastic organoids originate from single cancer stem cells, which undergo both self-renewal and differentiation, giving rise to stem and non-stem cancer cells, respectively. Therefore, organoids represent a precious tool to investigate the abnormal self-renewal and differentiation of cancer stem cells in order to identify new druggable targets. LCOs can be used for all of these applications (Figure 2), although their enormous potential is still underexploited. Actually, the majority of studies performed with LCOs have focused on drug screening, while basic aspects of lung cancer pathogenesis have been investigated by relatively few studies so far. Pioneering insights into the mechanisms of lung tumorigenesis were provided by two studies published in 2017 and in 2020 that used organoid systems to investigate the effects of specific gene mutations. First, Zhang et al. demonstrated that the sequential expression and activation of oncogenic KRAS and the deletion of Lkb1 were able to transform lung epithelial organoids into fully malignant organoids and promoted the transition towards a LUSC phenotype [46]. Later, Dost and coworkers used organoid systems derived from human iPSC and murine lung epithelial cells to model LUAD development. The expression of oncogenic KRAS in alveolar epithelial progenitor cells in both systems resulted in a change of transcriptional programs with downregulation of genes related to differentiation and maturation [67]. This was also the first study employing lung organoids to obtain transcriptional and proteomic profiles of normal epithelial progenitors as compared to early-stage lung cancer, providing a comprehensive molecular landscape of KRAS-driven lung tumorigenesis [67]. Starting in 2020, a handful of studies performed on LCOs provided new insights on the role of specific genes in lung cancer and/or on the biological functions of lung cancer cells. The oncogenic transformation of murine lung epithelial cells kept in organoid cultures was performed by Semba et al. using KRASG12V or EML4/ALK. Transformed organoids were then transplanted into the lungs of syngeneic mice, recapitulating the pathogenesis of human LUAD [68]. A model of tumorigenesis based on engineered mouse LCOs was recently employed to investigate the mechanisms of SCLC metastasis, finding that deficiency of the histone H3 lysine 4 methyltransferase KMT2C promoted extensive tumor metastasization [69]. Further insights in multi-step lung carcinogenesis were provided by Miura et al., showing that exogenous HER2 expression in iPSC-derived human lung organoids induced the formation of tumor-like structures and a transcriptional profile similar to LUAD with HER2 amplification [70]. Nacarino-Palma and colleagues investigated the role of Aryl Hydrocarbon Receptor (AHR) in murine lung organoids, showing that AHR loss cooperates with oncogenic KRAS to promote organoid stemness and self-renewal [71]. Sàndor and coworkers used LUAD organoids to identify a Wnt-producing microniche composed by both cancer cells and fibroblasts, which was able to influence cancer cell proliferation and extracellular vesicle release. The same authors identified a CD133high subpopulation in LCOs with stem cell characteristics, such as elevated self-renewal and Wnt activity or responsiveness [56]. Finally, Ma and coworkers analyzed the gene expression profiles of LCOs derived from LUAD and LUSC, finding a differential regulation of pathways involved in immune regulation, inflammation, MAPK, and Rap1 activation between the two histotypes [57]. In the upcoming years, studies on LCO basic biology are expected to increase, providing important insights into lung cancer therapeutic vulnerabilities.

### 4.2. LCOs for Modeling the Tumor Microenvironment: Reconstituted or Holistic Co-Cultures?

The TME includes normal epithelial cells, endothelial cells, mesenchymal-derived cells (fibroblasts and pericytes), and an immune cellular network composed by T and B lymphocytes, natural killer cells, macrophages, dendritic cells, mastocytes, and myeloid-derived suppressor cells. All of these cells communicate with cancer both through direct interactions and through the transfer of biologically active molecules contained in extracellular vesicles [72]. Tumor–TME interactions play a key role in supporting malignant proliferation, chemoresistance, immune escape, and metastatic spreading. Therefore, dissecting tumor–TME interactions through suitable co-culture platforms is essential to understand the biological mechanisms underlying all the stages of tumor development. Creating faithful models of the normal and neoplastic lung microenvironment is particularly important for several reasons: first, to allow a better understanding of lung cancer initiation upon chronic inflammation caused by chemicals, smoking, or air pollution; second, to identify new therapeutic strategies that target the pro-tumor and pro-metastatic effects of the TME; third, to study cancer immunosuppression and increase the efficacy of immune-based therapies [73]; and finally, to understand the properties of the lung microenvironment as a metastatic niche, since the majority of solid tumors metastasize to the lungs [74]. To this end, a complex bronchioalveolar lung organoid (BALO) system has been recently developed by culturing mouse bronchioalveolar stem cells with lung-resident mesenchymal stem cells and macrophages, thus providing new opportunities to study epithelial–immune–mesenchymal interactions in a normal lung in vitro model [75]. An adaptation of the BALO system to study cancer–TME interactions would likely provide new insights into the role of the lung microenvironment in supporting the growth of primary and metastatic tumors. Organoids are a promising tool used to model the interactions between tumor cells and cellular TME components (Figure 2) and have been previously co-cultured with cancer-associated fibroblasts (CAFs) or immune cells in several tumor settings [72,76]. Lung cancer cells organized as spheroids [77,78,79] or organoids [80] were co-cultured with CAFs, showing that the latter promote tumor cell proliferation and invasiveness. A systematic approach to LCO co-culture with immune cells was established by Dijkstra et al. and subsequently published as an independent protocol by the same group [43,44]. The authors performed co-cultures of autologous LCOs and peripheral blood lymphocytes from patients with NSCLC or colorectal cancer. LCOs were generated from six NSCLC patients and autologous tumor-reactive CD8+ T cells were obtained in two out of six cases [44]. Co-cultures of LCOs with autologous T cells can be used to dissect the mechanisms that determine tumor sensitivity or resistance to immunotherapy and possibly to generate patient-specific T cell products for adoptive T cell transfer. One step further in this direction was reported by Li H. and coworkers, who co-cultivated LCOs with CAR-T cells within a strategy to achieve T-cell-mediated targeting of NSCLC brain metastases [45]. Notably, organoid cultures in submerged Matrigel domes do not retain stromal or immune components of the primary tumor. Therefore, TME cells have to be isolated and added to established tumor organoid cultures, creating a so-called reconstituted TME model [76]. In an alternative approach, an air–liquid interface (ALI) co-culture system has been used to propagate organoids from lung and other tumors, preserving the endogenous tumor parenchyma, stroma, and embedded immune cells [47]. The ALI system is established starting from minced tumor fragments and has the advantage of maintaining the TME histological architecture and cellular complexity, thus representing a native (or holistic) TME model [76]. However, immune cells and fibroblasts cannot propagate long term in the ALI system, requiring further adjustment of this culture system in order to meet the necessities of different TME cell types.

### 4.3. LCOs in Translational Research: Applications for Personalized Medicine

In the last decade, the advent of molecularly targeted therapies and immunotherapies has brought significant advances in lung cancer treatment. However, therapeutic regimens are assigned according to broad categories, such as histological subtypes and/or genetic mutations, and the enormous potential of personalized medicine has yet to be accomplished. The implementation of personalized treatments for lung cancer is further complicated by the complexity of its genomic landscape. LUAD was shown to have higher rates of mutations as compared to other cancers, resulting in a large burden of passenger events that confounds the identification of actionable driver genes [81]. In addition, oncogenic driver mutations have been shown to be different among early and late disease stages [82]. Furthermore, even patient subgroups defined by the same oncogenic drivers show a substantial molecular diversity that results in heterogeneous clinical behavior and variable responses to anticancer therapies [82]. The reasons that limit the application of individualized cancer treatments include an insufficient availability of therapeutic response biomarkers, the impossibility to individually test therapeutic regimens before treatment, and a limited knowledge of the mechanisms responsible for intrinsic and acquired drug resistance. Patient-derived organoids (PDOs) provide an unprecedented opportunity to face these challenges and have been used in multiple tumor settings for drug screening and for exploring the effect of therapeutic candidates (reviewed in [83]). Concomitantly to the establishment of LCO technology in 2019, the first drug-testing experiments were performed to investigate whether LCO drug response was related to genomic alterations. Sachs et al. provided the first proof of principle that LCOs are amenable to drug screening, showing that LCOs largely retained driver mutations of parental tumors and displayed differential responses to chemotherapeutics and targeted inhibitors (i.e., erlotinib, gefitinib, crizotinib, and alpesilib) [34]. In the same year, Kim M. et al. established a biobank of 80 LCOs that recapitulated the histology and genetic features of major lung cancer subtypes and reported that LCOs responded to targeted drugs (i.e., olaparib, erlotinib, and crizotinib) according to their genomic alterations [40]. Takahashi and coworkers established a multi-tumor PDO biobank and investigated the response of three LCOs to a library of ~80 conventional and targeted drugs. Drug-testing experiments performed with LCOs were expanded to include three monoclonal antibodies (cetuximab, trastuzumab, and pertuzumab), one antibody-drug conjugate, and two immune checkpoint inhibitors (nivolumab and pembrolizumab, in experiments of T-cell-mediated killing), showing that LCOs are amenable to screening with different classes of compounds [41]. Further evidence to the feasibility of LCO-based drug screening was lately provided by four studies, all performed with rigorously validated LCOs. Shi and coworkers tested four LCO lines with three targeted inhibitors (trametinib, selumetinib, and afatinib) showing that organoid responses to targeted drugs were consistent with genetic mutations and PDX responses [17]. Chen et al. tested a library of ~20 compounds on three LCOs, confirming the correlation between the LCO response and the mutation profiles of parental tumors [37]. Li and coworkers generated a biobank of 12 LUAD LCOs that were shown to reproduce the histology, genomic landscape, and gene expression profile of parental tumors. LCOs were used for high-throughput drug screening with 24 compounds and for the identification of new tumor biomarkers associated with the survival status of LUAD patients [51]. Interestingly, this study reported that a number of drugs were effective on LCOs in the absence of the related mutation. On one side, this observation indicates that not 100% of organoids are recapitulative of the expected drug response, as previously observed in PDO-based clinical trials [84,85], and on the other side, it indicates that routine drug testing on LCOs may reveal patients that could unexpectedly benefit from targeted treatments [51]. An alternative approach to biomarker identification and selective pathway targeting was followed by Taverna et al., who used cytometry by time of flight (CyTOF) for lung cancer profiling. CyTOF analysis revealed tumors characterized by high AXL and JAK1 expression, epithelial-mesenchymal transition (EMT), and stemness markers, and the corresponding LCOs were sensitive to combined AXL/JAK1 inhibition [52]. In 2021, two studies provided significant insights in the field of LCO-based clinical modeling. Hu and coworkers established a method for rapid LCO generation from surgical resections or endobronchial cancer biopsies (with 79% and 40% success, respectively) followed by drug screening on a microwell array chip. The whole procedure of LCO production and testing took less than two weeks, being compatible with clinical needs [39]. Notably, Hu et al. avoided the overgrowth of normal lung organoids by using a medium devoid of factors required for the growth of normal lung cells, such as FGF7, FGF10, R-Spondin, and Noggin. Such culture conditions generated cancer-dominant cultures, although not 100% pure [39]. Importantly, the same study evaluated whether LCO-based drug sensitivity tests recapitulated patients’ responses to tyrosine kinase inhibitor (TKI) treatment, finding a positive agreement with clinical outcomes [39]. Finally, Kim S.Y. and coworkers established a biobank of 83 LCOs (of which 77 were from malignant effusions) from patients with advanced LUAD. LCOs largely retained driver mutations of parental tumors and, in a selected number of cases, recapitulated drug responses and progression-free survival of TKI-treated patients [35]. The same authors used LCOs to test new therapies against specific genetic alterations, reporting an efficacy of poziotinib against ERBB2 exon 20 insertions and pralsetinib against RET fusions [35]. Altogether, these studies have laid the ground for the use of LCOs in personalized medicine, encouraging further efforts to bridge the gap between preclinical and clinical research (Figure 2). Finally, clinical trials are currently ongoing in several countries with the aim of testing the correlation between LCOs sensitivity to treatments and clinical responses (NCT04859166 and NCT05092009 in the Netherlands, NCT03979170 in Switzerland, NCT03655015, NCT05332925 and NCT03896958 in the USA, NCT05136014 and NCT04826913 in France, and NCT03453307 and NCT03778814 in China). Altogether, the knowledge generated by LCO-based studies will pave the way for their future use as predictive models of treatment response.

## 5. Future Directions and Conclusions

### 5.1. Exploiting Biomaterial Technologies for an Improved Mimicking of the Lung ECM

The lung extracellular matrix (ECM) is a complex structure composed by at least 150 different proteins and by a variety of other insoluble extracellular macromolecules (glycoproteins, proteoglycans, polysaccharides, etc.) that serve as a binding interface for hundreds of secreted proteins, creating an information-rich dynamic environment [86]. The lung ECM also has peculiar biomechanical and physical properties, being subject to repetitive mechanical deformations and fluctuations in O_2_ pressure. All these conditions are extremely difficult to reproduce in vitro, representing a challenge for the culture of both normal and malignant lung organoids [87]. The ECM has been shown to influence all the hallmarks of cancer, critically influencing tumor progression, metastasis, and resistance to chemotherapy [88,89]. Therefore, the biochemical and biophysical conditions used to mimic ECM in organoid cultures are likely to influence not only the success of organoid generation but also the results of subsequent studies. Matrigel, a hydrogel derived from mouse sarcoma, is the preferred substrate used for LCO development (Table 1) due to its close resemblance to human ECM and its ability to promote cell adhesion with high efficiency. Matrigel is composed of four major basement membrane proteins and of a broad array of bioactive soluble proteins [90,91], thus providing key ECM signaling cues to tumor cells. However, Matrigel may not be the perfect substrate for an LCO culture for several reasons. First, Matrigel presents high batch-to-batch variability of protein content that may cause reproducibility issues in organoid cultures [91]. In addition to its poorly defined biochemical composition, Matrigel does not allow for the modulation of the physical properties of the culture (i.e., stiffness) and is not amenable to bioprinting procedures. Furthermore, Matrigel composition (60% laminin, 30% collagen IV, 8% entactin, and 2–3% perlecan) [92] does not reflect the composition of the lung connective tissue, which is mainly based on collagen I and III fibers [93]. A variety of natural and synthetic materials are being explored as alternatives to Matrigel for organoid cultures [8,92,94,95,96]. Briefly, alternative scaffolds for organoid cultures can be divided in natural and synthetic substrates [92]. Natural substrates mainly consist of hydrogels composed of polysaccharides, proteins, and animal-derived mixtures including alginate [97], chitosan [98], gelatin [96], collagen [54], silk fibroin [99], and hyaluronic acid [100]. Synthetic alternatives to Matrigel include polyethylene glycol (PEG) and its derivatives [92]. PEG hydrogels have been used to study matrix-related changes in a LUAD cell line, providing a proof of principle that key properties of cultured cells depend on matrix biochemical and physical properties [101]. While synthetic materials cannot recapitulate the complexity in composition, structure, and bioactivity of the natural ECM, their controlled fabrication together with modular composition provide high reproducibility and adjustability. Moreover, new advances in biomaterial engineering are moving towards an increasing capacity to faithfully recapitulate the heterogeneity of the tumor ECM [102]. The use of new biomaterials for LCO technology will likely benefit from the advances in 3D culture protocols for normal organoids from lung or other tissues [87,103]. In this regard, a recent report described the use of hyaluronic acid hydrogels for the generation and expansion of iPSC-derived lung alveolar organoids, achieving an increased homogeneity in organoid size and structure [104]. Improving the reproducibility of LCO cultures by controlling organoid size, shape, and structure is crucial for a broader implementation of LCO-based drug-screening protocols.

### 5.2. Potential Applications of New 3D Technologies to LCO Cultures

The development of new technologies for 3D cell cultures, such as microfluidic engineering and 3D bioprinting, are starting to meet organoid technology, providing new directions for integrating TME components and for improving the reproducibility of organoid-based drug screening [51,52,83,105], respectively. In particular, the application of microfluidic organ-on-chip technology offers new opportunities to control biochemical and biophysical parameters in organoid cultures, such as stiffness, shear stress, geometry, and the relative flow of nutrients or waste products, thus recreating a simplified version of the real microenvironment. The combination of 3D bioprinting and organoid technology will improve reproducibility and promote the standardization of protocols in drug testing [83,105]. Genome-editing technologies are already employed to target single genes in organoids and their application to LCOs will allow a better understanding of genotype–phenotype correlations and drug–genotype correlations. Finally, the integrated use of imaging techniques with 3D bioprinting, computer vision, and machine learning will provide new insights into the dynamic aspects of drug response and into tumor heterogeneity, as demonstrated in recent studies on colorectal PDOs [106].

## 6. Conclusions

Studies published in the last four years have provided evidence that LCOs can recapitulate the histology, genetics, and drug sensitivity of parental tumors, opening new perspectives for tumor biology and personalized medicine. However, future research in the LCO field will have to face several challenges in order to translate LCO technology into a real clinical benefit for lung cancer patients. Among other issues, it will be crucial to understand exactly to what extent LCOs are able to recapitulate parental tumors in terms of the response to different classes of anticancer therapies. Furthermore, future studies should improve the capacity of LCOs to represent an increasingly broad spectrum of clinically relevant variables, including different genetic mutations and uncommon tumor histotypes. Standardization of organoid production will also be a crucial issue for improving the reproducibility of experiments across different laboratories and the scalability of organoid cultures for clinical needs. To this end, the generation of living LCO biobanks coupled with standardized drug sensitivity tests and molecular profiling technologies will be crucial to support clinical decisions and the planning of clinical trials.

Finally, the landscape of cancer targets is continuously expanding to include not only tumor-intrinsic factors, such as mutated oncogenes, but also tumor-extrinsic factors, such as cell states and alterations in the TME’s cellular and acellular components [107]. In this scenario, LCOs represent a unique tool for integrating the expanding aspects of lung cancer complexity and translating preclinical research into therapeutic advances.

## Figures and Tables

**Figure 1 cancers-14-03703-f001:**
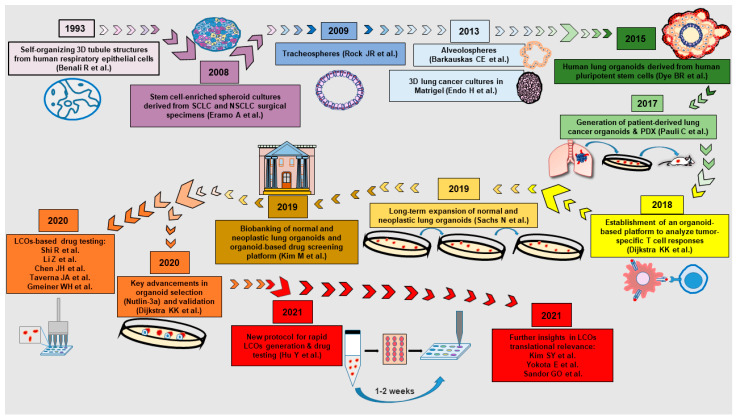
Timetable of landmark studies that have contributed to the generation of 3D cultures of lung epithelial cells. LCOs, lung cancer organoids; SCLC, small-cell lung cancer; NSCLC, non-small-cell lung cancer; PDX, patient-derived xenograft [9,16,17,22,23,24,25,27,32,34,35,37,38,39,40,42,44,58,59,60].

**Figure 2 cancers-14-03703-f002:**
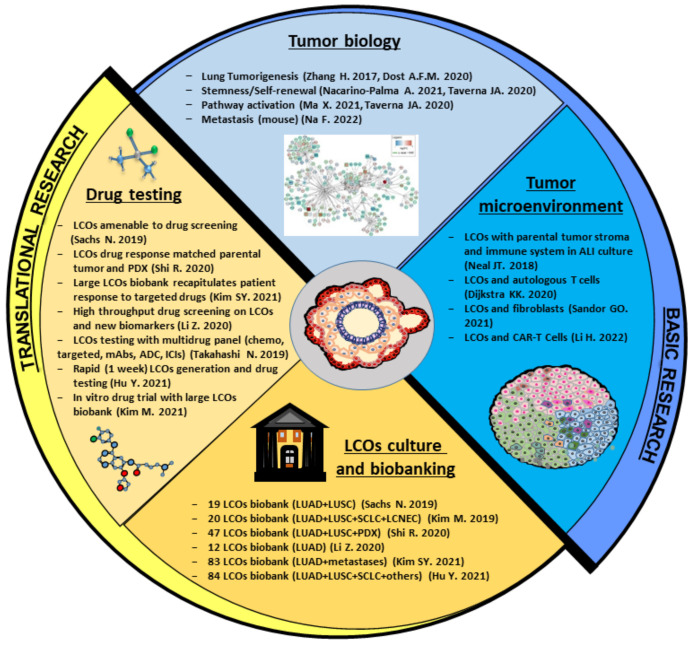
Applications of LCOs in basic and translational lung cancer research. TME, tumor microenvironment; LCOs, lung cancer organoids; ADCs, antibody-drug conjugates; ICIs, immune checkpoint inhibitors; ALI, air–liquid interface; LUAD, lung adenocarcinoma; LUSC, lung squamous cell carcinoma; PDX, patient-derived xenograft. Others: adenoid cystic carcinoma, sarcomatoid carcinoma, atypical carcinoid, mucoepidermoid carcinoma, large-cell neuroendocrine carcinoma. Not all the relevant references were reported in the figure due to space issues; we apologize to the authors who we were unable to cite [16,17,34,35,39,40,41,45,50,57,58,59,60,61,63,65,74].

**Table 1 cancers-14-03703-t001:** Main studies in the field of lung cancer spheroids and organoids. ECM, extra-cellular matrix; MBM, minimum basal medium; EGF, Epidermal Growth Factor; bFGF, basic Fibroblast Growth Factor; FGF, Fibroblast Growth Factor; HGF, Hepatocyte Growth Factor; LUAD, lung adenocarcinoma; LUSC, lung squamous cell carcinoma; LCNEC, large-cell neuroendocrine carcinoma; SCLC, small-cell lung cancer; NSCLC, non-small-cell lung cancer; TDS, tumor-derived spheroids; PDX, patient-derived xenograft; PDOs, patient-derived organoids; LCOs, lung cancer organoids; hESCs, human embryonic stem cells; GFR, growth factor reduced; NRG1, Neuregulin-1; IGF1, Insulin-like Growth Factor-1; DMEM/F12, Dulbecco’s Modified Eagle’s Medium and Ham’s F-12 Nutrient Mixture; N2, N2 supplement; B27, B27 supplement; XDOs, xenograft-derived organoids; ALI, air–liquid interface; SAG, smoothened agonist; PBLs, peripheral blood lymphocytes. Others: adenoid cystic carcinoma, sarcomatoid carcinoma, atypical carcinoid, mucoepidermoid carcinoma, LCNEC; NA, not available.

**SPHEROIDS**						
**YEAR**	**REFERENCES**	**ECM/SUPPLEMENTS**	**SOURCE**	** *N* **	**SUCCESS RATE**	**APPLICATIONS**
**2008**	[9]	**DMEM/F12 medium**, EGF and bFGF	Resection	7 lines of stem cell-enriched tumor-derived spheroids (TDS)	7/19 (36.8%)	Identification and characterization of lung cancer stem cells; generation of xenografts recapitulating the histology of parental tumors
**2013**	[22]	**StemPro hESC medium, Matrigel GFR**, NRG1, Long-IGF1, bFGF, Activin A, EGF	Resection/pleural effusion	108 TDS	Total: 108/143 (75.5%) 100/125 (80%) surgical samples, 8/18 (44.4%) pleural effusions	Method to expand patient-derived lung tumor cells
**2018**	[46]	**Advanced DMEM/F-12**, N2, Noggin, B27	Resection	3 TDS	100%	Method to expand patient-derived lung tumor cells
**2020**	[14]	**Collagen hydrogels in a 3D microfluidic culture system**	Core needle biopsy/surgical biopsy/pleural effusion	2 PDX-derived spheroids	2/2 (100%)	Drug testing
**ORGANOIDS**						
**YEAR**	**REFERENCES**	**ECM/SUPPLEMENTS**	**SOURCE**	** *N* **	**SUCCESS RATE**	**APPLICATIONS**
**2017**	[32]	**Matrigel GFR**, B27, N-acetylcisteine, R-spondin-1, Noggin, FGF10, FGF2, EGF, A83-01, Y-2763, SB202190, Nicotinamide, Prostaglandin E2	Resection/biopsy	1 PDO	1/2 (50%) LUAD	Biobanking
**2017**	[33]	**Advanced DMEM/F-12, Matrigel**, B27, N-acetylcysteine, Gastrin, Nicotinamide, EGF, Noggin, Wnt-3a, R-Spondin-1, A83-01, SB202190 and Y-27632	Resection/biopsy	3 PDOs	3/3 (100%) NSCLC	Evaluation of immune cell populations infiltrating cultured tissues; drug testing
**2019**	[34]	**Cultrex Basal Membrane Extract GFR**, R-Spondin-1, FGF7, FGF10, Noggin, A83-01, Y-27632, SB202190, B27, Nutlin 3a	Resection/biopsy	19 PDOs	Total: 19/34 (55.8%) 14/16 (87.5%) primary NSCLC 5/18 (27.8%) metastatic NSCLC	Long-term expansion of LCOs, validation, and drug testing
**2018**	[47]	**ALI**	Resection/biopsy	9 PDOs	9/20 (45%) NSCLC	New method for preserving endogenous tumor-infiltrating lymphocytes, suitable for immuno-oncology investigations and personalized immunotherapy testing
**2018**	[44]	**Geltrex Free Reduced growth factor basement membrane matrix**, B27, N-acetylcisteine, R-Spondin-1, Noggin, FGF10, FGF7, A83-01, Y-27632, SB202190, Nicotinamide, Nutlin-3a	Resection/biopsy	6 PDOs	6/6 (100%) NSCLC	Development of a platform to analyze tumor-specific T cell responses in a personalized manner
**2019**	[40]	**Matrigel GFR**, B27, N2, FGF2, EGF, Y-27632	Resection/biopsy	20 PDOs	Total: 20/23 (87%) 12/14 (85.7%) LUAD 5/6 (83.3%) LUSC 2/2 (100%) SCLC 1/1 (100%) LCNEC	Biobanking, drug testing
**2019**	[48]	**ECM base medium** without supplements	Resection/biopsy	1 PDO	NA	Investigation and inhibition of mitochondrial fission regulators in multiple tumor organoids
**2019**	[49]	**Matrigel**, B27, N2, FGF2, EGF, Y-27632	Resection/biopsy	NA	NA	Microfluidic platform-enabling LCO culturing and drug sensitivity tests
**2019**	[41]	**ECM base medium** without supplements	Resection/biopsy	3 PDOs	Total: 3/3 (100%) 2/2 (100%) LUSC 1/1 (100%) LUAD	Broad-spectrum drug testing
**2019**	[36]	**ECM base medium** without supplements	Pleural effusion	5 PDOs	5/5 (100%) LUAD	Establishment of an LCO culture system from pleural effusions; drug testing
**2019**	[50]	NA	Resection/biopsy	1 PDO	100%	Drug testing
**2020**	[17]	**Matrigel**, B27, N-acetylcysteine, Noggin, FGF10, FGF4, EGF, A83-01, Y-27632, CHIR 99021, SAG	Resection/PDX	19 PDOs 28 XDOs	Total: 47/65 (72.3%) 13/16 (81%) LUAD PDOs 9/13 (69%) LUAD XDOs 6/14 (43%) LUSC PDOs 19/22 (86%) LUSC XDOs (results are referred to short term LCO cultures)	Platform for LCO expansion and validation; drug testing
**2020**	[43]	**Geltrex**, B27, N-acetylcysteine, R-Spondin-1, Noggin, FGF10, FGF7, A83-01, Y-27632, SB202190, Nicotinamide	Resection/biopsy Autologous PBLs	NA	NA	Protocol for co-culture LCOs and autologous PBLs for the individualized testing of T-cell-based immunotherapy
**2020**	[16]	**Geltrex**, B27, N-acetylcysteine, R-Spondin-1, Noggin, FGF10, FGF7, A83-01, Y-27632, SB202190, Nicotinamide	Resection/biopsy	10 PDOs	10/58 (17%) (4 from primary tumor; 6 from metastasis)	Evaluation of several methods to identify tumor purity of organoids established from intrapulmonary tumors
**2020**	[51]	**Matrigel**, B27, N-acetylcysteine, R-Spondin-1, Noggin, FGF10, FGF7, A83-01, Y-27632, SB202190, Nicotinamide	Resection/biopsy	12 PDOs	12/15 (80%)	LCO biobanking and characterization; drug testing
**2020**	[37]	**MBM + Matrigel** (**1:3 ratio**), B27, N2, FGF2, EGF, Y-27632	Resection/biopsy	7 PDOs	Total: 7/7 (100%) 6/6 (100%) LUAD 1/1 (100%) LUSC	LCOs biobanking and characterization; drug testing
**2020**	[38]	**ECM base medium** without supplements	PDX derived from biopsies	4 XDOs	4/4 (100%) SCLC	Organoid generation from PDXs obtained from SCLC biopsies; drug testing
**2020**	[52]	**Matrigel**, B27, N2, R-Spondin-1, Noggin, FGF10, FGF2, EGF, A83-01, Y-27632, SB202190, Nicotinamide, Prostaglandin E2, HGF	Resection/biopsy	6 PDOs	6/11 (54.5%) LUAD	Testing of pathway inhibitors identified by single-cell proteomics
**2021**	[53]	**Matrigel**, B27, GlutaMAX, Noggin, FGF10, FGF7, SB202190, Nicotinamide, N-acetylcysteine, R-Spondin-1, Y-27632, A83-01	Resection/biopsy	12 PDOs	12/15 (80%) LUAD	Protocol for LCO generation from LUAD with high success rate
**2021**	[42]	**Matrigel**, B27, N-acetylcysteine, R-Spondin-1, Noggin, FGF10, FGF7, A83-01, Y-27632, SB202190, Nicotinamide, Nutlin-3a	Resection/pleural effusion	3 PDOs	Total: 3/41 (7%) 3/30 (10%) LUAD 0/7 (0%) LUSC 0/2 (0%) SCLC 0/2 (0%) Pleomorphic Carcinoma	LCO generation and characterization; targeted drug testing
**2021**	[35]	**Matrigel**, B27, N-acetylcysteine, R-Spondin-1, Noggin, FGF10, FGF7, A83-01, Y-27632, SB202190, Nicotinamide	Metastasis/pleural effusion	83 PDOs	83/100 (83%) LUAD	LCO generation and characterization; targeted drug testing
**2021**	[54]	**Matrigel**, B27, N2, Nicotinamide, N-acetylcysteine, Y-27632, EGF, SB202190, A83-01, Forskolin, Dexamethasone	Resection/biopsy	Refers to [39]		Method for on-chip LCO cryopreservation and drug testing
**2021**	[55]	**Matrigel**, B27, Y-27632, R-Spondin-1, Noggin, A83-01, Wnt-3a, EGF, FGF	Resection/biopsy	8 PDOs	8/10 (80%) SCLC	Generation and characterization of SCLC LCOs
**2021**	[56]	**Matrigel**, B27, N-acetylcysteine, R-Spondin-1, Noggin, FGF10, FGF7, A83-01, Y-27632, SB202190, Nicotinamide, Nutlin-3a, Heregulin-β1	Resection	6 PDOs	6/6 (100%) LUAD	Studies on cancer microniche and role of extracellular vesicles
**2021**	[39]	**Matrigel**, B27, N2, Nicotinamide, N-acetylcysteine, Y-27632, EGF, SB202190, A83-01, Forskolin, Dexamethasone	Resection/biopsy	84 PDOs	Total: 84/109 (77%) Resection: 55/71 (77.4%) LUAD 18/23 (78.2%) LUSC 4/4 (100%) SCLC 4/5 (80%) others Biopsy: 3/6 (50%) LUAD	Rapid LCO generation and drug testing by using a super-hydrophobic microwell array chip; consistency of in vitro results with clinical response
**2021**	[57]	**OmaStem Lung Cancer Medium**	Resection/pleural effusion	2 PDOs	2/6 (33.3%) LUAD	Differential gene expression analysis, prognostic analysis, and gene co-expression network analysis
**2021**	[45]	NA	Resection/biopsy	2 PDOs	NA	Drug testing (cisplatin sensitization by halofuginone)

**Table 2 cancers-14-03703-t002:** Major problems encountered in the generation and culture of lung cancer organoids and possible solutions. LCOs, lung cancer organoids; PDX, patient-derived xenograft.

MAIN PROBLEMS	POSSIBLE SOLUTIONS
**LOW SUCCESS RATE (overgrowth of normal airway organoids)**	Nutlin-3a Supplement (see Table 1) Growth factor deprivation of normal airway organoids [36,40,41] Extrapulmonary source such as pleural effusion or metastasis [34,35,36] Hand picking of LCOs [16,51] LCO generation from PDX obtained from primary tumor [17] or metastasis [38]
**LOW YIELD OF LCO CULTURES**	Pre-selection of tissue quality [16,34,40] Sample size of 1–4 cm^3^ [40,51] Practical tips: (a) Rapid processing (b) Digestion time established according to sample size [39,43,51] Culture medium optimization
**CULTURE EXTINCTION**	Mechanical dissociation during LCOs passaging [51]
**LCO GENERATION TOO LONG FOR CLINICAL APPLICATIONS**	Microwell array chip for rapid (1–2 weeks) LCO generation and drug testing [39]
**LCO PRODUCTION IS PARTICULARLY DIFFICULT FOR EARLY STAGE TUMORS**	Optimization of culture conditions [42]
**LIMITED HETEROGENEITY**	LCO generation from multiple biopsies from different areas of the same tumor or from both primary and metastatic tumors [16,42]

**Table 3 cancers-14-03703-t003:** Multi-step validations proposed to assess LCO purity. SCLC, small-cell lung cancer; NSCLC, non-small-cell lung cancer; H&E, hematoxylin/eosin staining; CK7, Cytokeratin 7; CK5/6, Cytokeratin 5/6; TTF-1, Thyroid Transcription Factor-1; CD56, cluster of differentiation 56, IHC, immunohistochemistry. The table reports information [16,40].

LCOs VALIDATIONS
		HISTOMORPHOLOGY	IHC	GENETIC	XENOGRAFT FORMATION
**NSCLC**	**ADENOCARCINOMA**	**Solid or cystic** 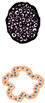	H&ENAPSINTTF-1CK7	COPY NUMBER LUNG CANCER-RELATED MUTATIONSWHOLE EXOME SEQUENCING 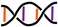	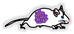
**ADENOSQUAMOUS CARCINOMA**	H&ECK7CK5/6p63
**SQUAMOUS CELL CARCINOMA**	H&Ep63TTF-1CK5/6
**SCLC**	**SMALL CELLCARCINOMA**	H&ECD56TTF-1Synaptophysin

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
