# Peer review of "Lung Cancer Organoids: The Rough Path to Personalized Medicine"

_cancers, 2022, doi:10.3390/cancers14153703_

Round 1
Reviewer 1 Report
This review is well-written and valuable for the researchers of cancer biology or tissue engineering.
However, there are some concerns that remain. Some sentences or descriptions should be added for the revision. The paper would be re-considered only when all the comments were responded to and reflected in a revised version.
The reviewer’ comments are below.
1.
New section
The biomaterial technologies for supporting 3D culture models should be briefly introduced by quoting related reviews and research papers. The biomaterial, such as Matrigel or gelatin can support the ECM components and assist the characteristics of 3D models. In the future, patient-derived cancer organoids will be combined with biomaterials.
I suggest at least these references to be added for the revision.
Overall for concept Cancers 2020, 12(10), 2754
Alginate Biomaterials 55 (2015) 110-118
Chitosan Biomaterials 25 (2004) 5147–5154
Gelatin  Tissue Eng. Part C Methods 2019, 25, 711–720 https://doi.org/10.1089/ten.tec.2019.0189
Collagen doi.org/10.1016/j.actbio.2018.06.003
Hyaluronic acid Adv. Healthcare Mater.2015, 4, 1664–1674
2.
The authors should examine the difference in results between cell line-derived spheroids and patient-derived organoids.
Author Response
This review is well-written and valuable for the researchers of cancer biology or tissue engineering. However, there are some concerns that remain. Some sentences or descriptions should be added for the revision. The paper would be re-considered only when all the comments were responded to and reflected in a revised version. The reviewer’ comments are below.
We are grateful to the Reviewer for his/her appreciation of the manuscript and for his/her helpful comments that allowed us to considerably improve the quality of our work.
- New section: The biomaterial technologies for supporting 3D culture models should be briefly introduced by quoting related reviews and research papers. The biomaterial, such as Matrigel or gelatin can support the ECM components and assist the characteristics of 3D models. In the future, patient-derived cancer organoids will be combined with biomaterials. I suggest at least these references to be added for the revision.
Overall for concept Cancers 2020, 12(10), 2754
Alginate Biomaterials 55 (2015) 110-118
Chitosan Biomaterials 25 (2004) 5147–5154
Gelatin Tissue Eng. Part C Methods 2019, 25, 711–720 https://doi.org/10.1089/ten.tec.2019.0189
Collagen doi.org/10.1016/j.actbio.2018.06.003
Hyaluronic acid Adv. Healthcare Mater.2015, 4, 1664–1674
We have added a new section (Section 5.1: Exploiting biomaterial technologies for an improved mimicking of the lung ECM) where we discussed the main issues related to biomaterials use in LCOs cultures. We cited the references indicated by the Reviewer and several other excellent studies in this field.
- The authors should examine the difference in results between cell line-derived spheroids and patient-derived organoids.
We discussed the difference in results between cell line-derived spheroids and patient-derived organoids at the beginning of the “Brief history” section (Section 2).
Reviewer 2 Report
This is a comprehensive and clear review of organoid developments in lung cancer. It details breakthroughs and ongoing challenges with these important preclinical models, suggesting future approaches for improving organoid recapitulation and scalability. This is a very topical area and, speaking as a lung cancer researcher (rather than an organoid one), I found the article very useful.
Suggestions for small improvements are:
1. 'Brief history' section. Using the word 'overgrown' - it took me a couple of times to appreciate what is meant by this. Rephrase or clarify.
2. 'Brief history' section. Clearly state difference between spheroids and organoids.
3. Table 1. Define XDO in legend.
4. Section 4.3. 'A number of drugs were effective on LCOs in the absence of the related mutation...'. It's suggested that this may reveal patients that could unexpectedly benefit from targeted treatments, but should this be qualified with some concern that these studies have produced organoids that are not recapitulative of the expected response?
5. The article would benefit from some analysis of why there is such large variation in success rates of organoids from different papers. Are there common patterns of success/failure?
6. Conclusions could offer a frank assessment of how far away we are from being able to offer real-time organoid models to patients. Also how far are we steering away from organoids being recapitulative for the purposes of making them viable (Nutlin etc)?
7. Replace 'advancements' with 'advances'. Overall the quality of English is very high.
Author Response
This is a comprehensive and clear review of organoid developments in lung cancer. It details breakthroughs and ongoing challenges with these important preclinical models, suggesting future approaches for improving organoid recapitulation and scalability. This is a very topical area and, speaking as a lung cancer researcher (rather than an organoid one). I found the article very useful.
We are extremely grateful to the Reviewer for his/her positive evaluation of our work and for raising several important issues, which we addressed in the revised version of the manuscript.
Suggestions for small improvements are:
- 'Brief history' section. Using the word 'overgrown' - it took me a couple of times to appreciate what is meant by this. Rephrase or clarify.
We have rephrased the sentence stating that “normal lung organoids tend to prevail over LCOs during the establishment of organoid cultures from primary lung tumors”.
- 'Brief history' section. Clearly state difference between spheroids and organoids.
We added a new paragraph at the beginning of the “Brief history” section where we discuss the differences spheroids and organoids and cite relevant references.
- Table 1. Define XDO in legend.
Done.
- Section 4.3. 'A number of drugs were effective on LCOs in the absence of the related mutation...'. It's suggested that this may reveal patients that could unexpectedly benefit from targeted treatments, but should this be qualified with some concern that these studies have produced organoids that are not recapitulative of the expected response?
Understanding to what extent PDOs are recapitulative of the expected clinical response is likely the most critical issue in organoid research. In our opinion, the answer is not straightforward. On one side, organoid technology has brought us one step further to reproducing many features of in vivo tumors as compared to previous preclinical models. On the other side, there are several evidences that PDOs are not 100% recapitulative of patients’ response to chemotherapy and targeted therapy, as shown by the TUMOROID and by the SENSOR clinical trials (Ooft SN 2019, Ooft SN 2021). The discrepancy between PDOs drug sensitivity and patients’ clinical response is likely due to a multiplicity of factors including the absence of a tumor microenvironment (that is known to crucially influence treatment outcome), the absence of vascularization, a different degree of heterogeneity between tumor and PDOs (this is particularly true for biopsy-derived PDOs) and the presence of poorly defined matrix components that may affect PDOs drug sensitivity. We have modified the sentence in Section 4.3 as follows: “Interestingly, this study reported that a number of drugs were effective on LCOs in the absence of the related mutation. This observation indicates on one side that not 100% of organoids are recapitulative of the expected drug response, as previously observed in PDOs-based clinical trials (Ooft 2019; Ooft 2021), on the other side that routine drug testing on LCOs may reveal patients than could unexpectedly benefit from targeted treatments”.
- The article would benefit from some analysis of why there is such large variation in success rates of organoids from different papers. Are there common patterns of success/failure?
We have analysed extensively the studies reporting a high versus low success in LCOs generation in the attempt to find common pattern of success or failure. We have considered several factors including tumor histotype, culture medium, source of material (primary tumor, metastases, pleural effusions) but we did not find a correlation with the success rate of LCOs generation. The only correlations that we were able to identify were between success rates and some technical variables related to LCOs production (i.e. size of the tumor sample, sample quality, processing and digestion protocols) that were already reported in Table 2. Finally, a possible correlation (of unknown significance) exists between high success rates and geographical origin of the studies, as many studies reporting high success in LCOs generation were from Eastern Asia (particularly China and Korea). This correlation may be due to an easier access to clinical samples by Eastern researchers and possibly to a larger size of tumor samples due to later diagnosis as compared to Western countries. However, this is only a speculation.
- Conclusions could offer a frank assessment of how far away we are from being able to offer real-time organoid models to patients. Also how far are we steering away from organoids being recapitulative for the purposes of making them viable (Nutlin etc)?
Thank you for your observations. We have modified the Conclusions accordingly.
- Replace 'advancements' with 'advances'. Overall the quality of English is very high.
We have done the substitution throughout the manuscript. Thank you.
Reviewer 3 Report
The authors have done a good job discussing the lung organoid development landscape and the various considerations that need to be made when working with such systems. However, the manuscript lacks good figures. The authors are requested to compile a figure from key references in the field demonstrating the different ways that lung organoids have been used or applied in the field of cancer. It will give the readers a visual aid when reading the manuscript.
Author Response
The authors have done a good job discussing the lung organoid development landscape and the various considerations that need to be made when working with such systems. However, the manuscript lacks good figures. The authors are requested to compile a figure from key references in the field demonstrating the different ways that lung organoids have been used or applied in the field of cancer. It will give the readers a visual aid when reading the manuscript.
We thank the Reviewer for his/her positive comments. We have reorganized Figure 2 in order to include a short description of main advances in the LCOs field together with relevant references.
Round 2
Reviewer 1 Report
.